# Simulation Analysis of Frog-Inspired Take-Off Performance Based on Different Structural Models

**DOI:** 10.3390/biomimetics9030168

**Published:** 2024-03-11

**Authors:** Shuqi Wang, Jizhuang Fan, Yubin Liu

**Affiliations:** State Key Laboratory of Robotics and System, Harbin Institute of Technology, Harbin 150001, China; wangshuqi@hit.edu.cn (S.W.); fanjizhuang@hit.edu.cn (J.F.)

**Keywords:** frog-inspired jumping robot, structural model, ADAMS simulation, jumping performance

## Abstract

The frog-inspired jumping robot is an interesting topic in the field of biomechanics and bionics. However, due to the frog’s explosive movement and large range of joint motion, it is very difficult to make their structure completely bionic. To obtain the optimal jumping motion model, the musculoskeletal structure, jumping movement mechanism, and characteristics of frogs are first systematically analyzed, and the corresponding structural and kinematic parameters are obtained. Based on biological characteristics, a model of the articular bone structure is created, which can fully describe the features of frog movement. According to the various factors affecting the frog’s jumping movement, mass and constraints are added, and the complex biological joint structure is simplified into four different jumping structure models. The jumping ground reaction force, velocity, and displacement of the center of mass, joint torque, and other motion information of these four models are obtained through ADAMS simulation to reveal the jumping movement mechanism and the influencing factors of frogs. Finally, various motion features are analyzed and compared to determine the optimal structural model of the comprehensive index, which provides a theoretical basis for the design of the frog-inspired jumping robot.

## 1. Introduction

The biological structure of natural organisms has gradually become reasonable over the course of long-term evolution, and the mode of movement has also demonstrated intelligence and adaptability [1,2]. The diversity of biological structures and their motions provides a steady stream of reference model samples for the bionic design and motion control of robotic structures [3,4,5,6]. Bionic jumping robots have gradually become a research focus in the field of robotics due to their unique advantages, which can show strong adaptability to different structural environments [7,8,9,10]. As a typical amphibian, the synergistic effect of different movement modes allows frogs to move efficiently and flexibly in various complex natural environments. Meanwhile, the intermittent amphibious motion mode helps improve the mobility and stability of the robot’s movement. Therefore, many researchers have studied bionic robots using frogs as objects [11,12]. The emergence of frog-inspired jumping robots based on a jumping mechanism has resulted in robots achieving better performance in crossing obstacles and handling complex terrain [13,14]. Its flexibility and adaptability enable robots to operate more efficiently in different environments, and it has potential practical application prospects. However, the current frog-inspired jumping robot still has defects, such as a large body shape, heavy weight, and a relatively complex structure, and there is no bionic design in the true sense. Therefore, it is of great significance to further analyze the biomechanical structure and jumping movement mechanism of frogs and reasonably simplify the structure to realize the miniaturization and the light weight of the robot.

The mechanism of frog movement is the basis and premise for research on frog-inspired robots. As the most basic method for analyzing the skeletal characteristics of frogs, the anatomical method was first used to study the muscles and bones of frogs [15,16]. Combined with the theory and method of biomechanics, biological information data and musculoskeletal characteristics can be obtained that form a database for the structural design and optimization of the frog-inspired robot [17,18,19]. Although the anatomical method can analyze the structural and joint movement characteristics from a biological perspective, it is not suitable as a conventional method for studying the movement mechanism due to its high cost and low universality. The most direct and effective means of obtaining frog movement data is to build an experimental observation platform [20,21]. Based on the collected movement data, the position information of each joint and torso during the frog movement is analyzed, its movement trajectory is extracted, and the corresponding velocity and acceleration are calculated to analyze its movement mechanism [22]. Although significant progress has been made in observing the jumping movement of frogs, there are still some challenges. One of them is the difficulty of observing living frogs in real time because their jumping movement is very short and explosive. Researchers need to continuously improve imaging techniques to obtain more accurate data and establish multiple suitable conversion coordinate systems for trajectory analysis [23]. In addition, the above two methods also have the disadvantage that the experimental variables are not controllable, and the repeatability is poor.

The development of simulation technology provides a new research method to study the movement mechanism. Bionic motion simulation based on the size of the biological structure and experimental data is the most convenient method for analyzing a frog’s motion mechanism. The simulation of the frog’s autonomous swimming based on computational fluid dynamics (CFD) was realized using the FLUENT software [24]. While the properties of the flow field structure in frog swimming were determined, the propulsion efficiency was also investigated using the control variable method. It can be seen that the simulation method has the advantages of low cost, strong operability, and good predictability. Therefore, the simulation analysis method is used to study the mechanism of frog jumping motion. As we all know, the frog’s jumping movement depends on its unique and efficient bone structure. The coordinated movement of the limbs allows the frog to jump quickly in a complex environment while displaying excellent agility. Researchers have begun to carry out frog movement mechanisms and frog-inspired robot structure design based on a bone structure model [25,26]. It is found that the existing structural models have the disadvantage of large size, which cannot effectively simulate the shape of the frog, and only one structural model is analyzed, which cannot reflect the influence of leg structure on the movement performance. Therefore, we established different frog bone structure models in ADAMS based on the actual biological size and analyzed the influence of the joint degrees of freedom (dof) on the movement performance of the robot through autonomous jumping simulation. This study not only helps to understand the jumping mechanism of frogs but also provides a reference for the structural design of frog-inspired robots and the research into other bionic robots.

In the following part, the biological characteristics of frogs were first studied in detail, including the composition and size of their musculoskeletal system, as well as the movement information when jumping. On this basis, virtual modeling and simulation analysis are carried out by adding attributes such as quality and constraints. The skeleton structure model that can fully describe the movement characteristics of frogs is established in Section 3. According to the number of dof, it can be divided into four structural models. The locomotion information of the four models is obtained by autonomous jumping simulation. Finally, the structural model with the best comprehensive index is determined by analysis and comparison, which provide a theoretical basis for the design of the prototype of the frog-inspired jumping robot.

## 2. Analysis of Frog Jumping Movement

### 2.1. Musculoskeletal Analysis

Analysis of the musculoskeletal properties of frogs is the basis of structural modeling. The frog’s body can be divided into head, torso, and limbs. The head is flat and pointed, which can reduce resistance when moving. The development of its limbs is not balanced; the forelimbs are short, and the hind limbs are long, and the quality is mainly concentrated on the torso. Taking the leopard frog as an example, its weight is about 28 g. The basic dimensions of its limbs are as follows: big arm (12.5 mm), small arm (12.5 mm), thigh (28 mm), calf (27.5 mm), tarsal bone (15 mm), and metatarsal bone and phalange (32 mm). The foot is long and flexible, and there are flippers between the toes. The musculoskeletal structure of the frog is shown in Figure 1a [6,27]. It can be seen that the skeleton mainly includes the skull, scapula, spine, and pelvis. The forelimb bone consists of the humerus, radius, wrist, and metacarpus bones. The shoulder joint consists of the scapula and humerus, which has three dofs. The elbow joint formed by the humerus and radius and the wrist formed by the radius and wrist bones each have only one dof. The hind limb bones can be divided into five parts: femur, tibia, fibula, tarsal bones, metatarsals, and phalanges. The femur is rod-shaped, and its end is spherical, which is connected to the acetabulum to form a hip joint similar to the spherical pair and has three dofs. The movements of the tibia and fibula relative to the femur, the movement of the tarsometatarsal bone relative to the tibia and fibula, and the movement of the metatarsophalangeal bone relative to the tarsal bone form knee joints, ankle joints, and tarsometatarsal joints with one dof.

Most of the energy required for the frog’s amphibious movement is generated by muscle contraction, particularly of the muscles of the hind limbs. The most important feature is the traction type, and the movement of the joint is realized by the corresponding muscular traction–skeletal system. As shown in Figure 1b, the semimembranosus, gluteus, biceps femoris, and gastrocnemius muscles play an important role in the extension and recovery of the hind limbs. The semimembranosus is a double-jointed muscle intertwined in the hip and knee joints and acts primarily on the hip joint; the action of the gluteus is similar to that of the biceps femoris, which mainly acts on the extension and contraction of the knee joint. The gastrocnemius muscle is a spring-shaped double-jointed muscle that extends through the aponeurosis of the foot in the knee and ankle joints, mainly acting on the ankle joint. It can be known that these muscles form an energy transfer system. Most muscles participate in more than one motor task, and different groups of interneurons produce movement patterns for each task [28]. The muscle is stimulated by bioelectric signals to produce a contraction when the frog is ready to take off. The immediate release of energy is transmitted to each joint, resulting in corresponding tension and explosive movement. This is the result of its flexibility and high adaptability to the control of the nervous system.

To further analyze the structural characteristics of frogs, the range of motion of each joint was observed through anatomical analysis [18]. First, all bones are placed in the same plane and the local coordinate systems for each joint are established, as shown in Figure 2a. The origin of the local coordinate system at the hip joint is on the acetabulum, with the positive *X*-axis pointing from the right to the left acetabulum. Place the origin of the knee joint local coordinate system at the junction of the femur and the tibia and fibula, with the positive *X*-axis parallel to the tibia and fibula. The local coordinate system of the wrist and tarsometatarsal joint is similar to that of the knee joint, with the coordinate origin at the junction of adjacent bones and the *X*-axis parallel to the bones. The *Z*-axis direction of all coordinate systems is perpendicular to the outward plane, and the *Y*-axis direction is determined by the righthand rule. The forms of movement of the frog joints are defined using the established local coordinate system. Rotation around the *Z*-axis is the main form of movement of the frog’s hind limbs. This is a flexion and extension movement. Each joint of the hind limbs has flexion and extension movements. Rotations about the *X*- and *Y*-axes only occur at the hip joint, which are considered abduction and adduction movements and internal and external rotation movements, respectively. The joint range of the frog is shown in Figure 2b, with the first column showing the three movements of the hip joint. The range of flexion-extension is −45°–90°, the range of abduction and adduction is 40°–140°, and the range of internal and external rotation is −50°–50°; the first row shows the range of flexion and extension of the knee joint, ankle joint, and tarsometatarsal joint. The corresponding movement angles are 0°–155°, −150°–0°, and −10°–140°, respectively. The range of these joint angles is the limit position that the joint can reach. These angular ranges are the maximum positions that the joints can reach, and the specific range of motion of the joints depends on the initial state and movement form of the jump.

### 2.2. Jumping Motion Information Analysis

According to the analysis of the frog jumping movement, it can be divided into three different stages: take-off stage, flight stage, and landing stage. The complete jumping process is shown in Figure 3 [25,27]. Before take-off, the frog’s flippers are fully open and touch the ground. Its thighs and calves are tight, and the muscles are tense. The forelimbs constantly adjust their posture to prepare for the jump. The frog’s movement from crouching to lifting the flipper from the ground is defined as the take-off stage. At this stage, the muscles of the hind limbs are stimulated to contract quickly and instantly expand to propel the frog away from the ground. The forelimb supports the torso when adjusting the departure angle. As the forelimb leaves the ground, they travel back to either side of the torso. During this period, the hip, knee, and ankle joints begin to move under the pull of the hind limb muscles, while the tarsometatarsal joint expands only slightly and most of the flippers are still in contact with the ground. When other joints reach a certain angle, the tarsometatarsal joint begins to extend rapidly so that the flippers gradually leave the ground, and the hind limbs are fully open and at approximately the same level as the torso. When the flippers completely leave the ground, the take-off stage ends and transitions to the flight stage. The frog is in the flight stage from the hind limbs leaving the ground and the forelimbs touching the ground again. The hind limbs continue to maintain the stretched state in the air, and the forelimbs gradually stretch forward from both sides of the torso, which can increase the moment of inertia, so that the frog can maintain the balance of posture and gradually change the direction of movement. When the direction of movement is toward the ground, the hind limbs begin to contract, and the forelimbs straighten and fully open the flippers in preparation for contact with the ground. The forelimbs touching the ground again means the flight stage ends and transitions into the landing stage. The landing stage is the process of the frog touching the forelimbs back to the ground to restore stability. At this stage, the coordinated movement of each joint of the forelimbs gradually lowers the center of gravity, and the hind limbs recover quickly. As the hind limbs fully recover and contact the ground, the center of gravity shifts backward, and the frog tends to be stable again. The landing stage is over to prepare for the next jump.

Since the frog’s mass is concentrated primarily in the torso, the centroid of the torso can be analyzed as the centroid of the mass. The trajectory model of the frog jumping movement is shown in Figure 4, where *P_i_* (*i* = 0, 1, 2, 3, 4) represents the position of the centroid at each stage. *P*_0_ represents the centroid position of the frog before take-off; *P*_1_ represents the centroid position when the flippers are just leaving the ground during the take-off stage; *P*_2_ represents the position of the centroid when the frog reaches its highest point during the flight stage; *P*_3_ represents the centroid position when its forelimbs touch the ground during the landing phase; *P*_4_ represents the centroid position of the frog after full landing. If air resistance is not taken into account, the frog is only affected by gravity during the flight stage, so its centroid motion can be viewed as a ballistic motion, that is, an oblique throwing movement [29]. Therefore, the trajectory of the frog’s centroid is a parabolic curve. *F*, *F*_0_, and *F*_1_ represent the combined force of the ground reaction force pointing toward the centroid, where *F*_0_ represents the ground reaction force when the frog begins to take off, and *F*_1_ represents the ground reaction force when the flippers are about to leave the ground, and *F* represents the ground reaction force at any point during the take-off stage.

The frog’s jumping speed can be decomposed into the horizontal and vertical speeds, which gradually increase in the take-off stage. When entering the flight stage, the horizontal speed remains basically unchanged, while the vertical speed decreases under the effect of gravity. Assuming that the frog passes through a time *t* during the flight stage, its centroid jumping distance is *S*, and its horizontal velocity is *v*, then there is *v = S*/*t*. Long-distance jumping is a prominent feature of frogs that is related to speed and time and is mainly expressed in the flight stage. Therefore, speed and jump distance can be used among the evaluation criteria for frog jump performance.

The magnitude and direction of the ground reaction force on the frog during take-off is constantly changing, but its direction is around the line between the toe contact point and the center of mass. Therefore, the line between these two points can be approximated as the direction of the ground reaction force. The ground reaction force is very low as the flippers begin to move during the take-off stage, is greatest in the middle stage, and rapidly decreases to zero upon exiting the ground. The overall change trend is similar to the sine curve change, which can effectively reduce the impact of the hind limb on the ground, prevent the reaction force from becoming too large so that the model leaves the ground prematurely, and extend the action time of the reaction force to improve jumping efficiency. It can be seen that the change in ground reaction force in the starting phase, as a key phase of the entire jumping process, affects jumping performance and, therefore, can be used as one of the criteria.

The key to frog jumping is that the muscles attached to the joints contract and release energy to create propulsive torque that stretches the joints. As a mediator of energy and joint extension, joint torque is the specific form of energy and the premise of joint extension and is also the key to achieving explosive movement. In addition, its size also provides a data basis for the design of the driving unit. Therefore, it is of great importance to analyze the driving torque of each joint, which can also be used an evaluation criterion. Determining the evaluation criteria for jumping performance lays the foundation for the subsequent selection of a suitable jumping structure model.

## 3. Numerical Modeling

### 3.1. Skeleton Structure Modeling

It can be seen from the above that the biological skeletal structure of the frog is very complex, and it is difficult to realize the bionic design of its structure using existing technical means. Therefore, it is of great importance to analyze the influence of biological skeletal structure on the jumping performance and establish a simplified model with similar functions to frogs, which can realize posture adjustments, take-offs, landings, and other actions. As shown in Figure 5, a frog skeletal structural model involving the torso, limbs, and joints is proposed, which is used to study the force generated by multiple joints and the intuitive influence of factors such as joints, muscles, and dofs on the frog’s movement.

The frog’s forelimbs mainly have three joints, namely the shoulder joint, the elbow joint, and the wrist. Since the forelimbs mainly play a role in adjusting the take-off posture and cushioning the landing during the jumping process, it can be simplified as a structure consisting only of the shoulder joint and the elbow joint. The flexibility of the hind limb joints is relatively large, which has a significant impact on jumping. There are four main hind limb joints, namely the hip joint, the knee joint, the ankle joint, and the tarsometatarsal joint. Therefore, the main structure of the model includes big arms, small arms, torso, thighs, legs, soles, and flippers, with dimensions essentially the same as those of biological frogs. Based on the skeletal structure model, set the material properties of each part (aluminum) and then configure their mass and inertia. The specific parameters are listed in Table 1. By changing the influencing factors in the model, forward jump simulation models of different structures can be established.

The biological skeletal structure model was assembled in SolidWorks and imported into ADAMS. The model is divided into 13 components. By adding constraints and quality attributes, a virtual prototype is then obtained as shown in Figure 6. The shoulder joint and elbow joint of the forelimb are simplified to only one dof. Because they are flexion and extension movements, the rotation pair constraint is adopted, which can meet the needs of adjusting the take-off posture and landing support in the jumping movement. The hind limbs mainly provide the power in the jumping movement, and the hip joint has three dofs, so a ball hinge pair is used for constraint. The knee and tarsometatarsal joint each have one dof, so the rotation pair is used to constrain; it should be noted that although the ankle joint only has one dof, it is necessary to add a passive constraint to the ankle joint to simulate the movement, considering the trend of relative torsion between the foot and leg during the take-off process. Therefore, a ball hinge pair constraint is used here.

According to Newton’s third law, contact between the flippers and the ground is required to generate friction, which in turn generates a ground reaction force that allows the model to perform a jumping motion. After establishing the skeleton model, the contact model and materials need to be added and set for motion simulation. The size of the friction coefficient has an important influence on the friction force. The friction coefficient not only depends on the structural features of the surface but also largely depends on the underlying properties of the materials used and the physical and chemical state of the outermost layer [30]. The flippers in the simulation model are made of rubber, which is used in the actual preparation, and the ground is made of wood material. We know that the coefficient of friction between rubber and wood is approximately between 0.5 and 0.7 [31]. Therefore, we first preliminarily determined the range of friction coefficient according to the material used. It should be noted that the tree frog can produce large friction due to its flippers so that it can complete the vertical upward crawling movement [32,33]. Then, based on the analysis of the functional components and attachment mechanism of tree frog toe pads, the influence of the shape and material of the tree-frog-inspired attachment structure on the adhesion and friction force is studied [34]. Finally, in order to generate sufficient ground reaction force for a smooth start of the model and improve the success rate of the simulation, the friction coefficient is set to 0.7 through adhesion and friction analysis in the biological system. In addition, to prevent limbs from penetrating the wood, the stiffness, maximum damping coefficient, and penetration depth were set. According to the actual situation, the contact between the limbs and the ground is added, and the specific parameters are shown in Table 2. The input to the simulation includes the length and mass of the torso and limbs, as well as the trajectory of each joint. It is worth noting that the joint trajectory in the simulation can be converted into a spline driving function to allow each joint to achieve a frog-inspired trajectory. The marker point is the center point on the bottom arc of the selected flipper. The result of the simulation includes the ground reaction force, joint torque, speed, and jumping distance of the skeleton model in the take-off stage.

### 3.2. Simulation of Multiple Structural Models

The forward jump simulation model was established on ADAMS to test the effect of structural models with different dofs on jumping performance. Based on the different dofs of joints, a total of four forward jump simulation models are established, as shown in Figure 7.

Model 1: The frog’s hip joint has three directions of movement when jumping, including flexion and extension, abduction and adduction, and internal and external rotation. In addition to flexion and extension, the ankle joint has passive movement that can follow the movement of the foot. The knee joint and the tarsometatarsal joint only have one direction of movement, namely flexion and extension. The shoulder joint and elbow joint of the forelimb only have flexion and extension. Therefore, Model 1 can fully simulate the joint motion of frogs with a total of nine dofs, as shown in Figure 7a.

Model 2: Model 1 analyzed the jumping performance of the frog structural model with three degrees of freedom in the hip joint. However, compared to the other two movements, the driving form of the internal and external rotation movement is complex, and its rotation angle during the jumping process is small. Therefore, Model 2 ignores the internal and external rotation movement of the hip joint based on Model 1 and maintains other forms of joint movement. It has a total of eight dofs, as shown in Figure 7b.

Model 3: By observing the frog’s jumping movement, it was found that the ankle joint and the tarsometatarsal joint are flexion and extension movements in the same direction. Combining them into one degree of freedom, removing the ankle flexion and extension motion, and retaining the tarsometatarsal joint are considered. Therefore, the ankle and the tarsometatarsal joints are combined into a flexion and extension movement in Model 3. It has a total of seven dofs, as shown in Figure 7c.

Model 4: To simplify the model structure as much as possible, it can be assumed that the joint has only one plane of movement, and the movement at other planes can be ignored. Therefore, the other forms of movement are restricted, and only the flexion and extension form of each joint, as well as the passive constraints on the ankle joint, are preserved. Model 4 is established based on Model 3 without considering the abduction and adduction of the hip joint, which has a total of six dofs, as shown in Figure 7d.

### 3.3. Simulation Analysis

The squat angle of Model 1, which simulates the entire joint jumping movement of frogs, is 20°. Based on the actual motion characteristics of the frog and the initial state of each joint in the model, motion is applied to each joint of the model, including 135° flexion and extension, 40° abduction and adduction, 30° internal and external rotation of the hip joint, 155° flexion and extension of the knee joint, 150° flexion and extension of the ankle joint, 70° flexion and extension of the tarsometatarsal joint, 40° flexion of the shoulder joint, and 60° flexion and extension of the elbow joint. Since the model structure is a rigid body, which cannot be as flexible as the actual movement of the frog, it is necessary to extend the time of the take-off stage appropriately while maintaining the trend of joint change. Therefore, the hip joint and knee joint move first, followed by movement of the ankle joint, for 0.1 s, and the tarsometatarsus joint begins to move after 0.15 s. The simulation time is set to 1 s, and the number of simulation steps is 1000 steps. Gravity (9.8 m/s^2^) was added to better simulate the real motion environment. The angle changes of each joint in the take-off stage are shown in Figure 8. It is worth noting that Models 2, 3, and 4 can also be obtained here because they are simplified on the basis of Model 1. According to the principle of the control variable method, Models 2, 3, and 4 are consistent with Model 1 except for the trajectory of the simplified joint.

Taking Model 1 as an example, the jumping movement process is obtained by motion simulation, as shown in Figure 9, which has three motion stages. The blue dashed line in the diagram represents the centroid trajectory of the model, which is similar to the trend obtained by the frog jump analysis. Both have a parabolic shape, which indirectly verifies the rationality of the simulation motion. Then, the simulation motion data are obtained by post-processing for comparative evaluation of motion performance. The performance changes can be clearly confirmed based on the centroid motion trajectories of the four models.

Since the take-off stage is a key part of the entire jumping process, it determines the height and distance of the jump and is also the main stage of energy consumption. Therefore, the jumping performance of the take-off stage is analyzed in detail. The reaction force generated by the contact between the flipper and the ground drives the movement of the model, and the change curve is shown in Figure 10a. The ground reaction force is low at the beginning of the take-off and gradually increases with the jump. As the tarsometatarsal joint begins to move, the ground reaction force gradually reaches the maximum value and rapidly decreases to zero before leaving the ground. The maximum value of the horizontal component of the reaction force at this stage is 0.5 N, and the vertical component is relatively large, with a maximum value of 0.8 N. The horizontal component of the center of mass velocity gradually increases, as shown in Figure 10b; it reaches the maximum value when the flipper is just off the ground, with a peak value of 1.06 m/s. It is worth noting that the velocity after 0.23 s is actually not a constant value, mainly due to the short duration. The frog model entered the flight stage after approximately 0.23 s. During the very short period of 0.02 s from 0.23 s to 0.25 s, the velocity of the model did not decrease significantly, so it seems to be a constant from the figure. It can also be seen from the jumping distance of each model that the velocity gradually decreases under the effect of gravity and enters the landing stage. The centroid displacement curve is shown in Figure 10c, and its trend is similar to the actual frog motion curve. The displacement of the centroid increases slowly at first, then increases rapidly when the flippers are just leaving the ground and reaches its maximum when the forelimbs touch the ground. The maximum jump distance of the model is approximately 0.34 m. The torque for each joint is shown in Figure 10d. The joint torque is mainly used to adjust the take-off posture to bring the forelimbs off the ground in the early stage of the jump and is concentrated in the later stage to achieve an explosive jumping movement. The maximum driving torque of the flexion and extension movement of the hip joint is 13.2 N·mm, which mainly provides the force for jumping forward. The joint torque of the abduction and adduction movement is 7.7 N·mm. The internal and external rotation movement is relatively small and is only 1.1 N·mm. The knee joint moves with the hip joint, and the joint torque is 2.4 N·mm. The driving torque of the ankle joint is 2.4 N·mm, and the driving torque of the tarsometatarsal joint is slightly larger than the ankle joint, which is 3.2 N·mm.

According to the principle of the control variable method, the motion exerted by each joint is consistent with Model 1, except that there is no internal and external rotation in Model 2. At this stage, the maximum vertical component of the reaction force is 0.73 N, and the maximum horizontal component is 0.48 N. Different from the small change in ground reaction force, the maximum horizontal component of centroid velocity of Model 2 is 0.83 m/s, which is 0.23 m/s smaller than that of Model 1. It can be seen from the centroid displacement curve shown in Figure 11c that the maximum jump distance is about 0.27 m, which is about 80% of Model 1. The joint torque of the model is shown in Figure 11d. When there is no internal and external rotation movement, the joint torque required for the other two movements of the hip joint becomes smaller. The flexion and extension movement driving torque is 9 N·mm, the abduction and adduction torque is 4.2 N·mm, the knee joint driving torque increases to 6.6 N·mm, and the ankle and tarsometatarsal joint torques are basically unchanged: 2.4 N·mm and 3.4 N·mm, respectively.

It can be found from the frog jumping movement that the ankle joint and tarsometatarsal joint are flexed and extended, and the direction is consistent. Therefore, it is considered that it is collapsed into one dof, and the motion of the ankle joint is removed based on Model 2, while the motion of other joints is left unchanged, so as to analyze the effects on jumping performance. The jump simulation motion effect of Model 3 is shown in Figure 12. The vertical component of the ground reaction force is increased compared to the first two models; the maximum value is 0.85 N, and the maximum value of the horizontal component is 0.43 N. When the flipper just left the ground, the horizontal component of the center of mass velocity reached a maximum of 1.05 m/s. When the model is in the flight stage, the centroid displacement increases rapidly and reaches its maximum after landing. The jump distance is about 0.28 m, which is slightly worse than Model 1 and slightly better than Model 2. The joint torque for movement is shown in Figure 12d. Due to the combination of ankle flexion and extension movements, the joint torque required for the tarsometatarsal joint is increased to 6.7 N·mm, the knee joint torque is reduced to 3.3 N·mm, and the joint torque of the hip joint is 12.5 N·mm and 5.2 N·mm, respectively.

To simplify the model structure as much as possible, it can be assumed that the hind limb joint has only plane motion, ignoring the motion at other planes and only maintaining the flexion and extension motion form of each joint. Therefore, without considering the abduction and adduction movement of the hip joint, Model 4 is established on the basis of Model 3, and then the influence of the minimum number of degrees of freedom on the jumping performance of the frog is judged. The simulation settings are also consistent with the above model, and the motion effect is shown in Figure 13. It can be seen that the ground reaction force components in both directions are reduced to a certain extent, namely 0.6 N and 0.28 N, respectively. The centroid velocity and displacement also decrease to 0.58 m/s and 0.14 m, respectively, which is essentially half of those of Model 3. The joint torque is shown in Figure 12d. Since the hip joint only has one degree of freedom, all the joint torques of Model 4 have increased. The torque for the hip joint is 13.2 N·mm, the torque for the knee joint is 4.8 N·mm, and the torque for the tarsometatarsal joint is 7.6 N·mm.

## 4. Result

From the above simulation analysis, the motion parameters of each model can be preliminarily obtained. In order to further intuitively analyze their advantages and disadvantages, the individual comparison of each index is then carried out to filter out the movement model with the best jumping performance.

### 4.1. Ground Reaction Forces Analysis

The ground reaction force was first compared and analyzed. It is known that the change in force is mainly concentrated in the later stage of the take-off stage, so the change in force during this period is mainly analyzed. Each ground reaction force component of the four models is plotted on a graph for comparison, as shown in Figure 14. For the horizontal component, the change trend of Model 1 is similar to that of Model 3, and the force is basically the same in the later stage of the jump. The difference is that the peak value of Model 1 is slightly larger than that of Model 3, and the change in Model 1 in the early stage of the jump is smoother than that of Model 3. The action time of Model 2 is shorter than those of Model 1 and Model 3, and the force change trend in the early stage of the jump is the same as Model 3, and the peak value is similar. Compared to the first three models, the value of Model 4 is small, the action time is short, and the change is gentle. For the vertical component, the action times of Model 1 and Model 3 are similar, and they start and end almost at the same time, but the change of Model 1 is gentler than that of Model 3, and the peak value is slightly smaller than that of Model 3. The action time of Model 2 is shorter than those of Model 1 and Model 3, and the peak value is smaller than that of Model 1. When Model 2 reaches the peak value, Model 3 continues to increase. Model 4 has the shortest action time and the smallest vertical component peak.

As we know, the longer the duration of the total force, the more beneficial the extension of the frog’s hind limbs, and the more power it provides for the frog to jump. Therefore, Model 1 and Model 3 have the best effect on the ground reaction force, followed by Model 2. The ground reaction force of Model 4 is the smallest and begins to change first. When the hind limbs are not fully extended, they have been vacated, resulting in most of their energy not being used in the take-off stage, so the movement effect is at its worst.

### 4.2. Jumping Speed and Distance Analysis

Compare the horizontal velocity and jump distance of the centroids of the four models on the same graph, as shown in Figure 15. It can be seen that when the force value is maximum, the horizontal velocity of the center of mass also reaches the maximum value. The speed of Model 4 is the largest in the early stage of take-off, followed by Model 3, and Model 1 is similar to Model 2, so the acceleration of Model 4 is the largest. As the movement progresses, the speed of Model 4 reaches a maximum of 0.7 m/s at around 0.17 s, while the speeds of the other models continue to increase. The acceleration of the model increases rapidly at this time, and the speed reaches a maximum in a short time. The speeds of Model 1 and Model 3 are the highest at approximately 1.05 m/s. The speed of Model 2 is slightly larger than that of Model 4, which is about 0.8 m/s. The separate analyses of Model 3 and Model 1 show that the speed of Model 3 in the take-off stage is larger than that of Model 1, and the speed change is more stable.

The jump distance is mainly realized in the flight stage, and it can be seen that the jump distance hardly changes in the take-off stage and the landing stage. Model 4 takes off the earliest, but its shortest flight time and minimum speed lead to the shortest jump distance of about 0.14 m. In contrast, the jump distance of Model 1 is the largest; not only is the speed the largest, but the flight time is also the longest, about 0.6 s. The speed of Model 3 is similar to that of Model 1, but the short flight time makes the jump distance smaller than that of Model 1. Although the flight time of Model 2 is longer than that of Model 3, its speed is less than that of Model 3; therefore, the jump distance of Model 3 is relatively large. Based on previous analyses, we know that the flight stage can be approximated as a parabolic motion and the flight time is related to the jump height. The higher the jump height, the longer the time. Therefore, we can know that Model 1 has the highest jump height, followed by Model 2. Model 3 is slightly smaller than Model 2. Model 4 has the smallest jump height. Combined with the jumping speed, it is finally obtained that the jump distance of Model 1 is the farthest, followed by Model 3. Model 2 is smaller than Model 3, and Model 4 has the shortest jumping distance.

### 4.3. Joint Torque Analysis

The peak value of the joint torque obtained by each model is listed in Table 3, and the jump performance of the model is judged by analyzing the required torque of each joint to realize the jump motion.

The joint torques of Model 1 and Model 2 are first compared. Model 1 has internal and external rotational motion of the hip joint, which is a complex form of motion that affects the motion of other planes, resulting in greater torque required for flexion and extension motion and abduction than in Model 2. However, it does not have a significant impact on the motion of other joints, and the torque required for other joints remains basically unchanged. Next, a comparative analysis was conducted between Models 2 and 3. Model 3 incorporated the ankle joint into the tarsometatarsal joint, resulting in an increase in the required joint torque for the tarsometatarsal joint, which also affects other joints, and the joint torques are increased. Model 4 lacks three motions compared to the first three models, which increases the joint torque required for the remaining joint movements. The torque required for a single joint movement is the largest, and the driver required is the highest. But from the perspective of the hip joint, the joint torque of abduction is greater than that of other joints, so it requires the maximum external drive. Among the four models, Model 2 requires the smallest moment for hip joint flexion and extension motion, while the other three joint torques are all greater than 10 N·mm. Additionally, Model 2 also requires the smallest driving force due to the smaller joint torques. The torque required by each joint of Model 3 is similar to that of Model 4, but it is one more joint torque than in Model 4, so the drive of Model 4 is less than that of Model 3. Among all models, Model 1 requires the most joint torque and is the most complex.

### 4.4. Jumping Performance Analysis

The degree of freedom of each joint does not exist independently, its role is not only affected by other degrees of freedom; it will also affect the role of other degrees of freedom. According to the motion parameters, it can be known that the ground force of Model 1 is large, the action time is long, the speed and jump distance are the best, and it has good motion performance. However, it requires the most joint torque, and the joint motion is complex, with nine degrees of freedom, which requires high performance of the structure and is difficult to be realized by a mechanical structure. Model 2 with eight degrees of freedom requires less single-joint torque and stable motion, but its jump distance and speed, ground reaction force, and force action time are less than those of Model 1 and Model 3. The number of joints torques required by Model 4 is less than those of other models, but the single torque is the largest. Its ground reaction force is small, and the action time is short, and the jump height and jump distance are the smallest. The only advantage is that the number of degrees of freedom is the least. The single joint torque required by Model 3 is higher than that of Model 2, but the required joint torque is relatively small. The ground reaction force has a long action time, and the horizontal component and vertical component are balanced. The speed and jump distance are not greatly reduced compared with Model 1, and its degree of freedom is seven, which is less than those of Model 1 and Model 2. In summary, Model 3 is the optimal jump model.

## 5. Discussion and Conclusions

The frog jumping motion with different degrees of freedom is realized in this study, and the optimal jumping structure model is discussed in combination with simulation analysis. Since biological characteristics are the basis of bionic design, the composition of the musculoskeletal system and the mechanism of jumping movement of frogs are first studied in depth. The forelimb mainly plays the role of adjusting the movement posture and landing buffer during the jumping process. Strong hind limbs and pull-type muscle characteristics are the key to achieving explosive movement. On this basis, a simplified model with similar functions to frogs has been established, including the torso, limbs, and joints, which can realize posture adjustment, take-off, landing, and other actions. According to the number of degrees of freedom, four virtual prototypes are established by adding attributes such as mass and constraints, and the degree-of-freedom characteristics of the hip joint are obtained by using the control variable method. The simulation results show that the motion efficiency of the joint with nine degrees of freedom is the highest, and that of the joint with six degrees of freedom is the lowest. The motion performance of eight degrees of freedom is similar to that of seven degrees of freedom, but the control of the former is more complicated. Through the analysis of the motion characteristics of the ground force, velocity, and displacement of the four models, an optimal seven degree-of-freedom jumping motion model is determined, which will provide a theoretical basis for the design of the frog-inspired robot.

A summary of existing models compares and analyzes them in terms of size, mass, degrees of freedom, etc., as shown in Table 4. On the basis of analyzing a specific structure separately, we established four skeletal models by changing the degrees of freedom of the hip and ankle joints and analyzed the influence of different degrees of freedom on the motion performance. The simulation results are extracted for comparative analysis. It is found that the maximum hip joint torque of the six-bar mechanism model is 1.35 × 10^4^ N·mm, which is about 1000 times the flexion and extension motion torque of the hip joint of Model 4 [26]. From the perspective of ground reaction force and motion performance, it takes nearly 50 N to achieve a jumping distance of 1200 mm [25]. Such a large force is bound to put forward higher requirements for the driving unit, energy transfer, and robot structure [27]. In sharp contrast, our model can achieve a jump distance of 360 mm with only 0.85 N, which greatly improves motion efficiency. In addition, compared with the size and weight of the model, our structure is closer to the actual frog. The data obtained from the simulation analysis from the biological point of view are more valuable, which greatly improves the research efficiency and reduces the research cost. However, the influence of the multi-structural model on the motion performance is only analyzed by jumping simulation. This only provides a theoretical basis for the structural design of the robot from the perspective of simulation, and it also needs to be designed in detail according to the actual driving unit and motion requirements. Therefore, it is very necessary to carry out structural design experiments on this basis, and then, feedback to structural optimization simulation is also one of the future work contents. In addition, due to the take-off stage having a great influence on the whole movement process, we only focus on this stage. It should be noted that the stability of the frog landing stage is also a key research stage of robot design and motion control. Therefore, we will further study and analyze the whole motion process of the frog on this basis. It is worth noting that the animal–robot interaction model shows that age has a great influence on the range of motion of the hind legs. Older animals will produce early responses as strategic behaviors to compensate for slower muscle responses and limited exercise capacity, which is consistent with other compensation strategies we know [35]. The establishment of the interaction model between frogs of different ages and their jumping performances has important reference value for further study of jumping mechanisms. Last but not least, the external environment, such as whether the terrain is rugged and the degree of softness and hardness, can affect the motion performance of the frog. Therefore, in the process of structural design and selection of driving units, careful consideration should also be given to the usage environment of the robot.

## Figures and Tables

**Figure 1 biomimetics-09-00168-f001:**
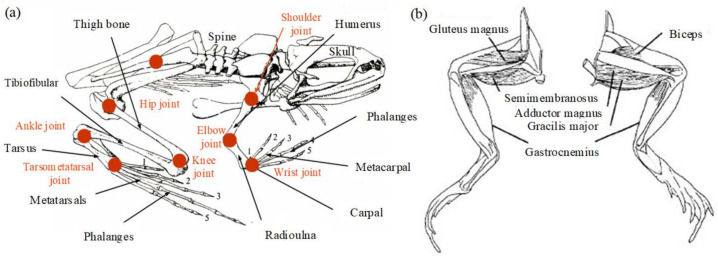
The musculoskeletal system of the frog. (**a**) The skeletal system and joints of the frog. (**b**) The muscular system of the hind limbs.

**Figure 2 biomimetics-09-00168-f002:**
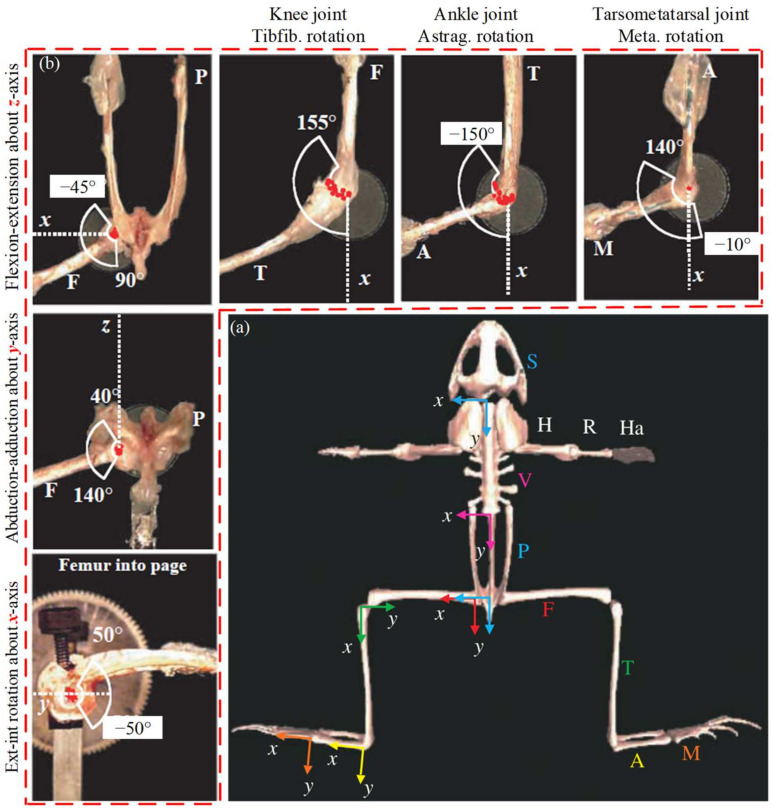
Anatomical analysis of frogs. (**a**) Establishment of skeletal coordinate system. (**b**) Range of motion of each joint.

**Figure 3 biomimetics-09-00168-f003:**
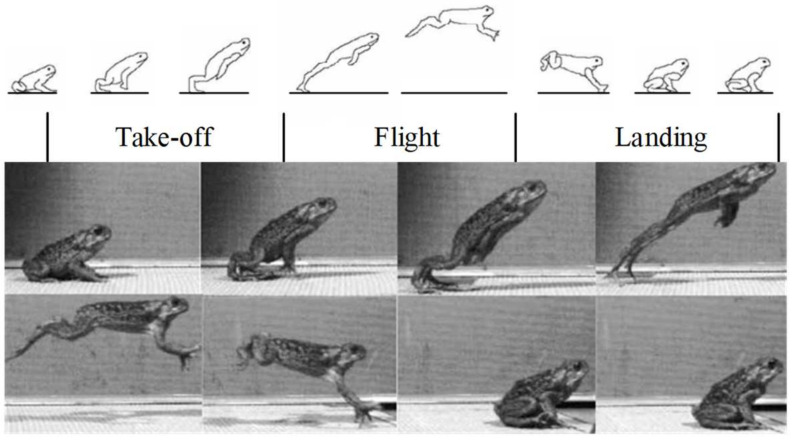
The frog jumping process diagram and the division of jumping stages, including the take-off, flight, and landing stages [25,27].

**Figure 4 biomimetics-09-00168-f004:**
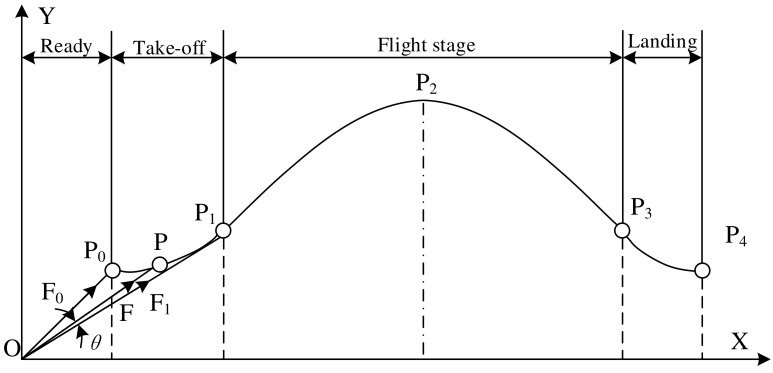
The analytical modeling principle of the jumpers: the propelling process, the approaching process, the interface rupturing process, and the falling deceleration process in air.

**Figure 5 biomimetics-09-00168-f005:**
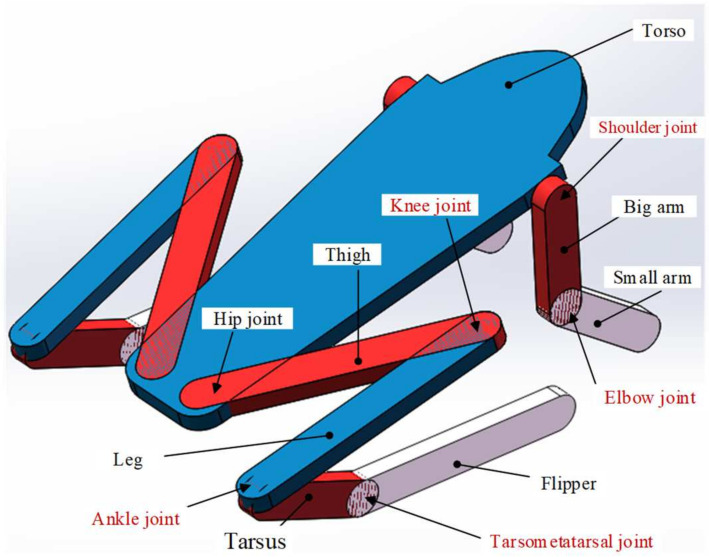
The biological skeletal structure model of a frog includes the trunk, limbs, and joints.

**Figure 6 biomimetics-09-00168-f006:**
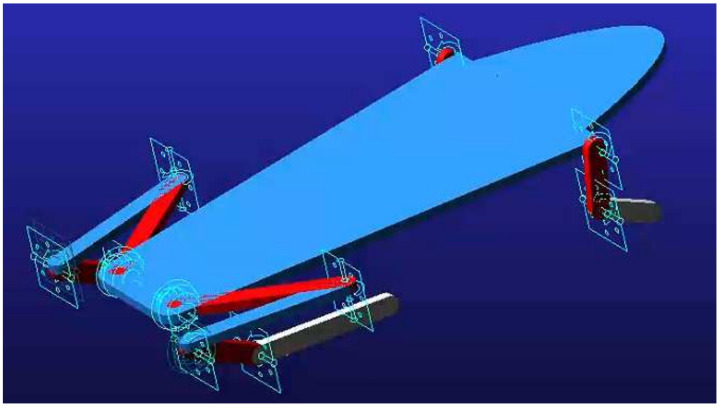
The frog biological skeleton model and its constraints in ADAMS.

**Figure 7 biomimetics-09-00168-f007:**
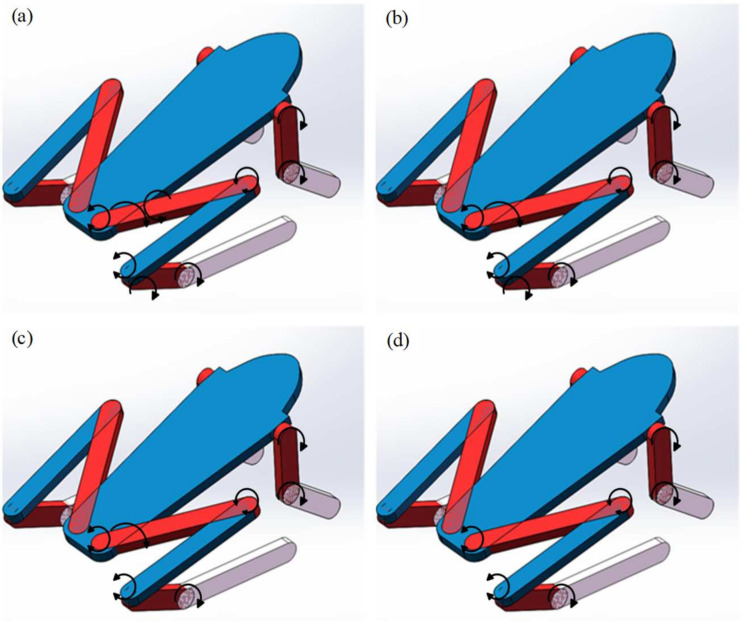
Four frog structure models with different dofs. (**a**) Model 1: The hip joint has three dofs (nine dofs in total). (**b**) Model 2: The hip joint has two dofs (eight dofs in total). (**c**) Model 3: The hip joint has two dofs, combining the ankle joint and the tarsometatarsal joint (seven dofs in total). (**d**) Model 4: The hip joint has one dof (six dofs in total).

**Figure 8 biomimetics-09-00168-f008:**
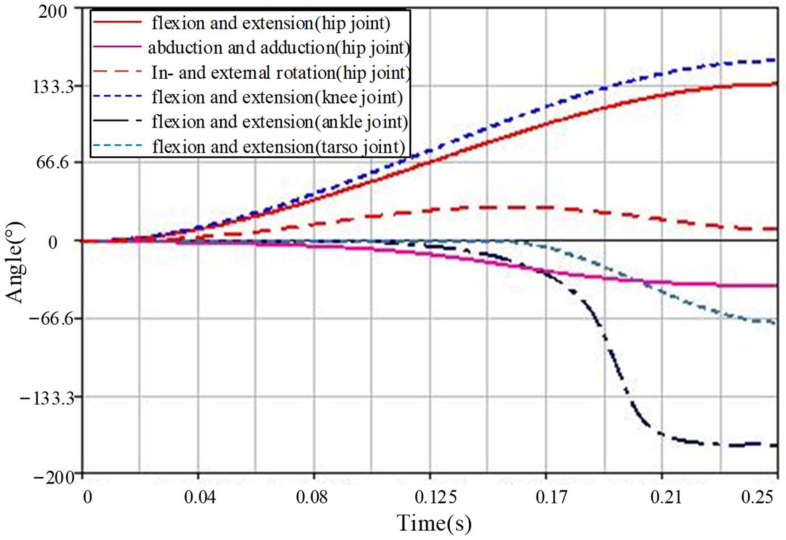
The trajectories of each joint during the take-off stage of Model 1.

**Figure 9 biomimetics-09-00168-f009:**
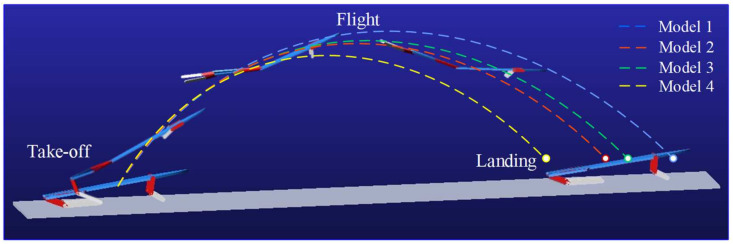
The simulation jumping motion processes and centroid trajectories of the four models.

**Figure 10 biomimetics-09-00168-f010:**
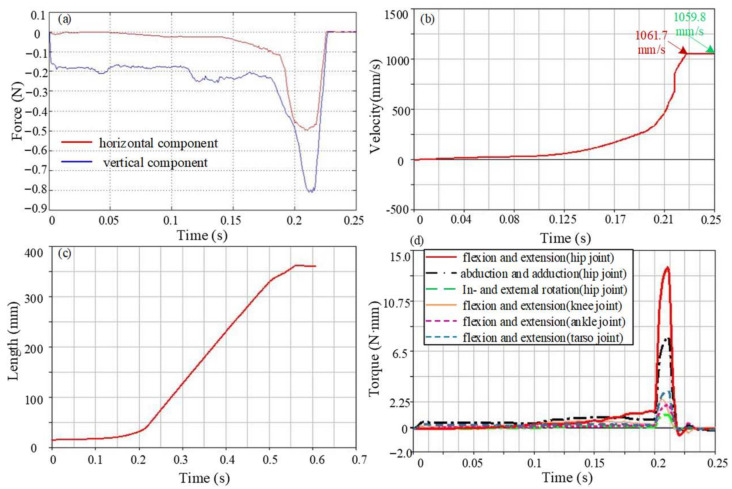
The motion performance parameter curve of Model 1. (**a**) The ground reaction force in the take-off stage, (**b**) Centroid velocity during motion, (**c**) Centroid displacement during motion, (**d**) Torque of each joint.

**Figure 11 biomimetics-09-00168-f011:**
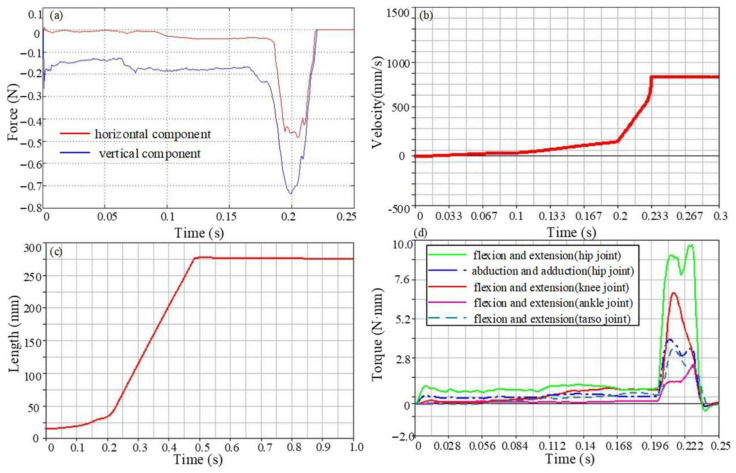
The motion performance parameter curve of Model 2. (**a**) The ground reaction force in the take-off stage, (**b**) Centroid velocity during motion, (**c**) Centroid displacement during motion, (**d**) Torque of each joint.

**Figure 12 biomimetics-09-00168-f012:**
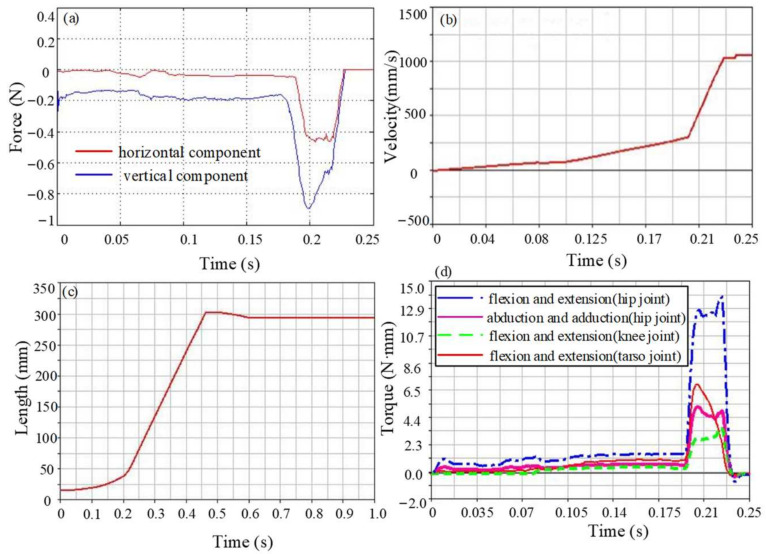
The motion performance parameter curve of Model 3. (**a**) The ground reaction force in the take-off stage, (**b**) Centroid velocity during motion, (**c**) Centroid displacement during motion, (**d**) Torque of each joint.

**Figure 13 biomimetics-09-00168-f013:**
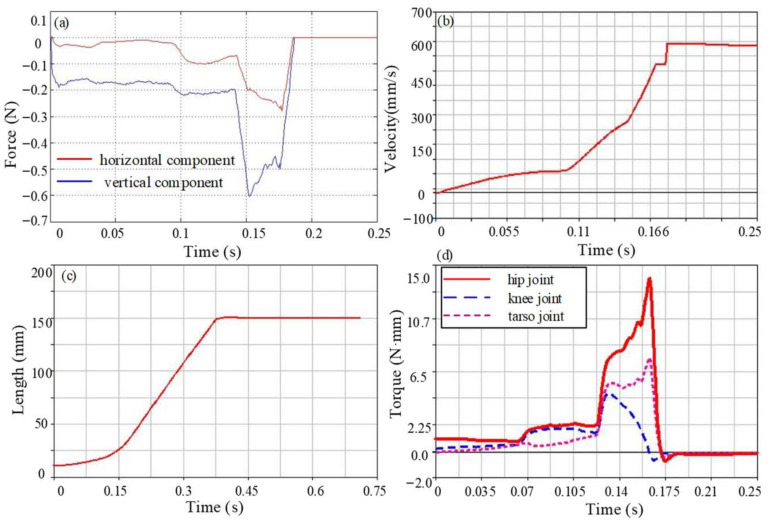
The motion performance parameter curve of Model 4. (**a**) The ground reaction force in the take-off stage, (**b**) Centroid velocity during motion, (**c**) Centroid displacement during motion, (**d**) Torque of each joint.

**Figure 14 biomimetics-09-00168-f014:**
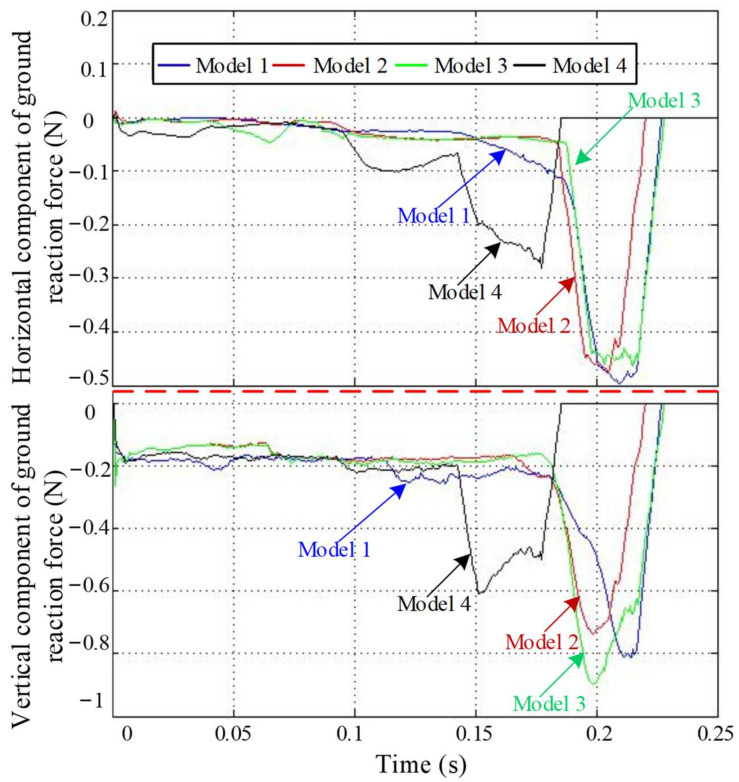
The component comparison diagram of ground reaction force in two directions.

**Figure 15 biomimetics-09-00168-f015:**
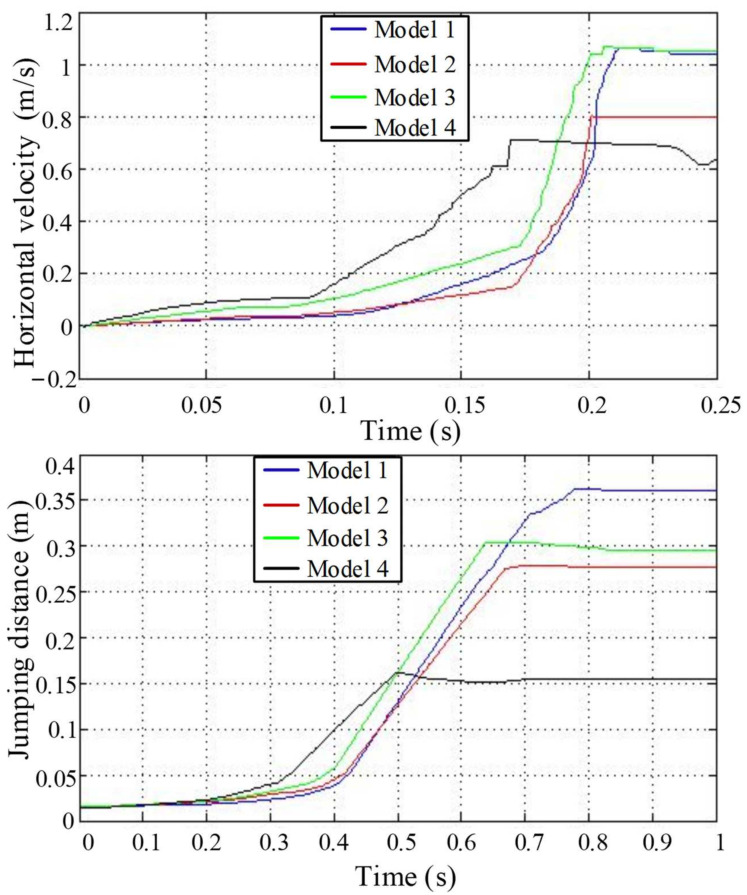
The component comparison diagram of jump speed and distance.

**Table 1 biomimetics-09-00168-t001:** The structural model of a frog and its corresponding parameters.

Model	Torso	Big Arm	Small Arm	Thigh	Leg	Sole	Flipper
Dimension (mm)	40 × 16 × 20	12.5	12	28	27.5	15	25
Mass (g)	20	0.5	0.5	1	1	0.5	0.5
Inertia (kg·m^2^)	5.6 × 10^−6^	1.8 × 10^−9^	1.5 × 10^−9^	2.1 × 10^−8^	2.0 × 10^−8^	3.1 × 10^−9^	8.3 × 10^−9^

**Table 2 biomimetics-09-00168-t002:** The parameters configuration of contact model.

Name	Stiffness	Damping	Force Exponent	Penetration Depth	Friction Coefficient	Stiction Transition
Value	1.0 × 10^7^ N/m	3.0 × 10^4^ N·s/m	2.2	1.0 × 10^−4^ m	0.7	0.1 m/s

**Table 3 biomimetics-09-00168-t003:** The torque of different joints of each model.

Model	Flexion and Extension (H)	Abduction and Adduction	Rotation	Flexion and Extension (K)	Flexion and Extension (A)	Flexion and Extension (T)
1	13.2	7.7	1.1	2.4	2.4	3.2
2	9.0	4.2	0	2.4	2.4	3.4
3	12.5	5.2	0	3.3	0	6.7
4	13.3	0	0	4.8	0	7.6

**Table 4 biomimetics-09-00168-t004:** Comparison of results from different simulation models.

Models	Size	Weight	Dofs	Jump Analysis	Simulation	Experiment	Motion Efficiency
Xu [26]	Big	Heavy	4	no	yes	yes	low
Zhong [27]	Big	Heavy	5	yes	no	yes	low
Wang [28]	Big	Heavy	9	yes	no	yes	low
This work	Small	Light	6–9	yes	yes	no	high

## Data Availability

The datasets generated during simulation or analyzed during the current study can be obtained from the corresponding author according to reasonable requirements.

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
