# Peer review of "Simulation Analysis of Frog-Inspired Take-Off Performance Based on Different Structural Models"

_biomimetics, 2024, doi:10.3390/biomimetics9030168_

Round 1
Reviewer 1 Report (Previous Reviewer 1)
Comments and Suggestions for Authors
Authors addressed several comments I provided.
Maybe they can introduce also an important aspect in jumping that is control and nerobehavioral processes that contribute in nature to exploit this locomotion strategy and that can inspire biomimetic artifacts. In this Journal I think these issues are relvant and shoudl be included.
Some examples
Romano, D., Bloemberg, J., Tannous, M., & Stefanini, C. (2020). Impact of aging and cognitive mechanisms on high-speed motor activation patterns: evidence from an orthoptera-robot interaction. IEEE Transactions on Medical Robotics and Bionics, 2(2), 292-296. DOI: 10.1109/TMRB.2020.2977003
Pearson, K. G. (1993). Common principles of motor control in vertebrates and invertebrates. Annual review of neuroscience, 16(1), 265-297.
Many parts of the manuscript are very difficult to understand. Please, address a careful language editing.
Comments on the Quality of English Language
English still need a careful revision.
Author Response
Dear Editor:
We are very grateful to your time and effort for improving our manuscript entitled "Simulation analysis of frog-inspired take-off performance based on different structural models" (No. 2864880). In the revised manuscript, we took all of the advices by doing a careful revision (Modified part plus blue). Detailed replies to the questions are presented as follows.
Response to Reviewer #1:
Thank you for your affirmation and valuable comments of this manuscript.
- Thank you for your suggestions and references. Indeed, as you say, control and neurobehavioral processes in nature help to exploit this locomotion strategy and can inspire bionic artifacts. Therefore, we have read, analyzed and cited the references you provided, and introduced the important aspects of jumping in Section 2.1 and Discussion, and revised and added the content of the relevant part as follows. Thanks again for your careful review and valuable comments.
The second paragraph of Section 2.1(Line 128-133): “Most muscles participate in more than one motor task and different groups of inter-neurons produce movement patterns for each task [36]. The muscle is stimulated by bioelectric signals to produce a contraction when the frog is ready to take off. The im-mediate release of energy is transmitted to each joint, resulting in corresponding tension and explosive movement. This is the result of its flexibility and highly adaptability to the control of the nervous system.”
The second paragraph of Conclusion (Line 590-599): “It is worth noting that the animal-robot interaction model shows that age has a great influence on the range of motion of the hind legs. Older animals will produce early responses as strategic behaviors to compensate for slower muscle responses and limited exercise capacity, which is consistent with other compensation strategies we know [39]. The establishment of the interaction model between frogs of different ages and their jumping performance has important reference value for further study of jumping mechanism. Last but not least, the external environment, such as whether the terrain is rugged and the degree of softness and hardness, can affect the motion performance of the frog. Therefore, in the process of structural design and selection of driving units, careful consideration should also be given to the usage environment of the robot.”
- Thank you for your suggestions on language. We have carefully read and used Paraphrasing Tool-Academic Rephrase Tool for Researchers revised the language of the manuscript (https://www.ref-n-write.com/paraphrasing-tool). Please review it again. Thank you again for the time and effort you have put into improving this manuscript.
If there are any more questions, please don’t hesitate to contact us at the address below. We would be grateful if the manuscript could be considered for publication in Biomimetics. Thanks again for your time and detailed comments.
Best wishes,
Yours sincerely
Corresponding author:
Name: Yubin Liu
E-mail: liuyubin@hit.edu.cn

Reviewer 2 Report (Previous Reviewer 2)
Comments and Suggestions for Authors
The response to the reviewer was well prepared. And in the revised manuscript, the issues pointed out in the previous review were well supplemented.
In the next stage of research, it would be good to present the results of actual fabrication and experimental evaluation.
Author Response
Thank you for your affirmation and valuable comments of this manuscript.
- Thank you for your good suggestion. We will carry out actual manufacturing and experimental evaluation in the next stage of research, with a view to presenting better results on the basis of this simulation study. Thank you again for your valuable suggestions.

Reviewer 3 Report (Previous Reviewer 3)
Comments and Suggestions for Authors
Authors have addressed some of the proposed comments in the previous review. Although there are some improvements included in the revised manuscript, my recommendation is that the presented research should be matured and expanded to be published in Biomimetics (JCR-Q1).
Comments for improving this version of the paper:
· Authors should justify the friction coefficient used of 0.7. The justification is weak, without any references or experimental basis. If authors used a contact model, please, define it in more detail (force law, markers, how the selected element works).
· Have authors added gravity force to the model? The Velocity (i.e. Figure 9) reaches a constant value of 1000 mm/s. Now, at least in one (Figure 9), add another figure increasing time in order to see how this velocity decrease. In the paper, the line is horizontal.
· Figure 6 should be improved from readability point of view. There are texts superposed. Not changed.
· Discussion should include comparison with the results of other authors. Not compared with results of the references. Which is the improvement of this research in comparison with previous published research?
Author Response
Dear Editor:
We are very grateful to your time and effort for improving our manuscript entitled "Simulation analysis of frog-inspired take-off performance based on different structural models" (No. 2864880). In the revised manuscript, we took all of the advices by doing a careful revision (Modified part plus blue). Detailed replies to the questions are presented as follows.
Response to Reviewer #3:
Thank you for your valuable comments on improving the quality of this manuscript.
- Response to comment: Authors should justify the friction coefficient used of 0.7. The justification is weak, without any references or experimental basis. If authors used a contact model, please, define it in detail.
Response: Thank you for your suggestions on the simulation parameters. In order to ensure that the structural model can take off smoothly, we add contact between the limbs and the ground. The frictional force generated by contact is a prerequisite for the ground reaction force and the key to ensuring that the model can take off. (Newton's Third law of Motion) Therefore, we have explained the rationality of using the friction coefficient 0.7 based on the materials used (rubber and wood) and provided a detailed definition of the contact model, so that readers can have a clearer understanding of the simulation parameters. The specific revises are as follows:
The fourth paragraph of Section 3.3 (Line 285-302): “According to Newton's third law, contact between the flippers and the ground is required to generate friction, which in turn generates a ground reaction force that al-lows the model to perform a jumping motion. After establishing the skeleton model, the contact model and materials need to be added and set for motion simulation. The flippers in the simulation model are made of rubber, which is used in the actual preparation, and the ground is made of wood material. We know that the coefficient of friction between rubber and wood is approximately between 0.5 and 0.7 [38]. In order to generate sufficient ground reaction force for a smooth start of the model and im-prove the success rate of the simulation, the friction coefficient is set to 0.7. In addition, to prevent limbs from penetrating the wood, the stiffness, maximum damping coefficient, and penetration depth were set. According to the actual situation, the contact between the limbs and the ground is added, and the specific parameters are shown in Table 2. The input to the simulation includes the length and mass of the torso and limbs, as well as the trajectory of each joint. It is worth noting that the joint trajectory in the simulation can be converted into a spline driving function to allow each joint to achieve a frog-like trajectory. The marker point is the center point on the bottom arc of the selected flipper.”
Table 2. The parameters configuration of contact model.
Name |
Stiffness |
Damping |
Force Exponent |
Penetration Depth |
Friction coefficient |
Stiction Transition |
Value |
1.0E+07N/m |
3.0E+04 N·s/m |
2.2 |
1.0E-04 m |
0.7 |
0.1m/s |
- Response to comment: Have authors added gravity force to the model? The Velocity (i.e. Figure 9) reaches a constant value of 1000 mm/s. It is necessary to explain.
Response: Thanks for your careful review on detail. We added gravity to better simulate the real motion environment. It is worth noting that the velocity is actually not a constant value, mainly due to the short duration. The frog model entered the flight stage from about 0.23s. During the very short period of 0.02s from 0.23s to 0.25s, the speed of the model did not decrease much, so it seems to be a constant value from the figure. We marked the speed values of two points in the Figure 10. It can also be seen from the jump distance of each model that the speed will gradually decrease under the action of gravity and enter the landing stage. Thank you again for your careful review. To avoid similar confusion for readers, we have provided additional explanations in the manuscript as follows:
The first paragraph of Section 3.3 (Line 350-351): “Gravity (9.8m/s2) was added to better simulate the real motion environment.”
The third paragraph of Section 3.3 (Line 380-385): “It is worth noting that the velocity after 0.23s is actually not a constant value, mainly due to the short duration. The frog model entered the flight stage after approximately 0.23s. During the very short period of 0.02s from 0.23s to 0.25s, the velocity of the model did not decrease significantly, so it seems to be a constant from the figure. It can also be seen from the jumping distance of each model that the velocity gradually de-creases under the effect of gravity and enter the landing stage.”
The Figure 10 (Line 399-400):
Figure 10. The motion performance parameter curve of model 1.
- Response to comment: Figure 6 should be improved from readability point of view. Not changed.
Response: Thank you for your careful review on Figure 6. Figure 6 shows the established frog skeleton structure model and the relevant constraints added in Adams. In order to enhance the readability and clarity of Figure 6, we have made improvements as follows (Line 283-284):
Figure 6. The frog biological skeleton model and its constraints in ADAMS.
- Response to comment: Discussion should include comparison with the results of other authors. Not compared with results of the references. Which is the improvement of this research in comparison with previous published research?
Response: Thank you for your suggestions for the discussion and conclusion. We have added Table 4 for comparison and analysis with other research results, and cited the relevant research results in the references. On this basis, the limitations of this study are summarized, which points out the direction for future work. Thanks again for your good suggestion. Specifically, we revised the Discussion as follows: (Line 571-577):
The second paragraph of Discussion (Line 572-578):
Table 4. Comparison of results from different simulation models.
Models |
Size |
Weight |
Dofs |
Jump analysis |
Simulation |
Experiment |
Xu [33] Zhong [34] Wang [35] This work |
Big Big Big Small |
Heavy Heavy Heavy Light |
4 5 9 6/7/8/9 |
no yes yes yes |
yes no no yes |
yes yes yes no |
Summarize the existing models and compare and analyze them in terms of size, mass, degrees of freedom, etc., as shown in Table 4. From the comparison results, it can be seen that the size and mass of the structural model are closer to the actual frog, the influence of different degrees of freedom on take-off performance has also been analyzed based on different structural models, which greatly improves the research efficiency and reduces the research cost [34-35].
If there are any more questions, please don’t hesitate to contact us at the address below. We would be grateful if the manuscript could be considered for publication in Biomimetics. Thanks again for your time and detailed comments.
Best wishes,
Yours sincerely
Corresponding author:
Name: Yubin Liu
E-mail: liuyubin@hit.edu.cn

Round 2
Reviewer 3 Report (Previous Reviewer 3)
Comments and Suggestions for Authors
Authors have addressed some of the proposed comments in the previous review. Although there are some improvements included in the revised manuscript, my recommendation is that the presented research should be matured and expanded to be published in Biomimetics (JCR-Q1)
Comments for improving the paper:
· Authors should justify the friction coefficient used of 0.7. The justification have been improved , however is not still sound .
· Discussion should include comparison with the results of other authors. Not compared with results of the references. Which is the improvement of this research in comparison with previous published research?
Author Response
Dear Editor:
We are very grateful to your time and effort for improving our manuscript entitled "Simulation analysis of frog-inspired take-off performance based on different structural models" (No. 2864880). In the revised manuscript, we took all of the advices by doing a careful revision (Modified part plus blue). Detailed replies to the questions are presented as follows.
Response to Reviewer #3:
Thank you for your recognition of the manuscript revision and your valuable comments improving the quality of this manuscript. We have made a detailed supplement according to your comments, please review again.
- Response to comment: Authors should justify the friction coefficient used of 0.7. The justification has been improved, however is not still sound.
Response: Thank you for your suggestions on the simulation parameters. The frictional force is the key to ensuring that the model can take off smoothly and the completion of the motion simulation. The size of the friction coefficient has an important influence on the friction force. The friction coefficient depends not only on the structural features of the surface, but also to a large extent on the underlying properties of the materials used and the physical and chemical state of the outermost layer [1]. Therefore, we first preliminarily determined the friction coefficient range of 0.5-0.7 according to the material used, because the friction coefficient between rubber and wood is in this range; Then, we analyzed the size of the friction force generated by the frog in the process of movement through the cited references, especially the friction force generated by a tree frog that can move vertically in the process of climbing upward [2-3]. Based on the analysis of the functional components and attachment mechanism of tree frog toe pads, the influence of the shape and material of the tree frog-inspired attachment structure on the adhesion and friction force is studied [4]. Finally, the final friction coefficient was determined by the adhesion and friction analysis in the biological system.
We hope that this explanation will enable readers to have a clear understanding of the reasonable choice of friction coefficient. Thanks again for your variable comment. The specific revises are as follows:
[1] Barnes WJ, Goodwyn PJ, Nokhbatolfoghahai M, Gorb SN. Elastic modulus of tree frog adhesive toe pads. J Comp Physiol A Neuroethol Sens Neural Behav Physiol. 2011 Oct;197(10):969-78.
[2] Meng FD, Liu Q, Wang X, et al. Tree frog adhesion biomimetics: Opportunities for the development of new, smart adhesives that adhere under wet conditions. Philosophical Transactions of the Royal Society A: Mathematical, Physical and Engineering Sciences,2019,377(2150).
[3] Hanna G, Barnes W-J P. Adhesion and detachment of the toe pads of tree frogs. Journal of Experimental Biology,1991,155:103-125.
[4] Julian K A Langowski, Dimitra Dodou, Peter van Assenbergh, Johan L van Leeuwen. Design of Tree-Frog-Inspired Adhesives, Integrative and Comparative Biology, Volume 60, Issue 4, October 2020, Pages 906–918
The fourth paragraph of Section 3.3 (Line 284-302): “According to Newton's third law, contact between the flippers and the ground is re-quired to generate friction, which in turn generates a ground reaction force that allows the model to perform a jumping motion. After establishing the skeleton model, the contact model and materials need to be added and set for motion simulation. The size of the friction coefficient has an important influence on the friction force. The friction coefficient not only depends on the structural features of the surface, but also largely depends on the underlying properties of the materials used and the physical and chemical state of the outermost layer [31]. The flippers in the simulation model are made of rubber, which is used in the actual preparation, and the ground is made of wood material. We know that the coefficient of friction between rubber and wood is approximately between 0.5 and 0.7 [32]. Therefore, we first preliminarily determined the range of friction coefficient according to the material used. It should be noted that the tree frog can produce large friction due to its flippers so that it can complete the vertical upward crawling movement [33-34]. Then, based on the analysis of the functional components and attachment mechanism of tree frog toe pads, the influence of the shape and material of the tree frog-inspired attachment structure on the adhesion and friction force is studied [35]. Finally, in order to generate sufficient ground reaction force for a smooth start of the model and improve the success rate of the simulation, the friction coefficient is set to 0.7 through adhesion and friction analysis in the biological system.”
- Response to comment: Discussion should include comparison with the results of other authors. Not compared with results of the references. Which is the improvement of this research in comparison with previous published research?
Response: Thank you for your suggestions for the discussion and conclusion. Sorry for didn't really understand your meaning before. We made a Table 4 to compare and analyze the different characteristics of the models in the references and this manuscript. The simulation results of each model are extracted, and the joint torque, ground reaction force and jump distance are compared and analyzed. The motion efficiency is added as an evaluation criterion to Table 4. On this basis, we further summarize the advantages of our model and compare the results in detail to reflect the progress of this research. Thanks again for your good suggestion. Specifically, we revised the Discussion as follows:
The second paragraph of Discussion (Line 583-597):
Table 4. Comparison of results from different simulation models.
Models |
Size |
Weight |
Dofs |
Jump analysis |
Simulation |
Experiment |
Motion efficiency |
Xu [26] Zhong [27] Wang [28] This work |
Big Big Big Small |
Heavy Heavy Heavy Light |
4 5 9 6-9 |
no yes yes yes |
yes no no yes |
yes yes yes no |
low low low high |
A summary of existing models and compare and analyze them in terms of size, mass, degrees of freedom, etc., as shown in Table 4. On the basis of analyzing a specific structure separately, we established four skeletal models by changing the degrees of freedom of the hip and ankle joints, and analyzed the influence of different degrees of freedom on the motion performance. The simulation results are extracted for comparative analysis. It is found that the maximum hip joint torque of the six-bar mechanism model is 1.35×104 N.mm, which is about one thousand times of the flexion and extension motion torque of the hip joint of Model 4 [27]. From the perspective of ground reaction force and motion performance, it takes nearly 50 N to achieve a jumping distance of 1200 mm [26]. Such a large force is bound to put forward higher requirements for driving unit, energy transfer and robot structure [28]. In sharp contrast, our model can achieve a jump distance of 360 mm with only 0.85 N, which greatly improves the motion efficiency. In addition, com-pared with the size and weight of the model, our structure is closer to the actual frog. The data obtained from the simulation analysis from the biological point of view are more valuable, which greatly improves the research efficiency and reduces the research cost.
If there are any more questions, please don’t hesitate to contact us at the address below. We would be grateful if the manuscript could be considered for publication in Biomimetics. Thanks again for your time and detailed comments.
Best wishes,
Yours sincerely
Corresponding author:
Name: Yubin Liu
E-mail: liuyubin@hit.edu.cn

This manuscript is a resubmission of an earlier submission. The following is a list of the peer review reports and author responses from that submission.
Round 1
Reviewer 1 Report
Comments and Suggestions for Authors
The manuscript titled "Simulation Analysis of Frog-Inspired Jumping Performance Based on Different Structural Models" by Wang et al. provides an analysis of the biomechanics and bionics involved in designing a frog-inspired jumping robot. The study aims to understand the optimal jumping motion model by analyzing the musculoskeletal structure, jumping motion mechanism, and motion characteristics of frogs. The authors establish a joint bone structure model based on biological characteristics and simplify it into four different jumping structure models, considering various factors affecting frog jumping motion.
The work presents several crucial issue that makes this study not suitable for publication in the current status.
In addition, the manuscript could benefit from improved clarity in presentation. The authors should consider revising sentences or providing additional explanations to enhance the overall readability of the document.
Authors should enhance the clarity and coherence of the introduction.
The state of the art on jumping inspired robots is very poor and authors should expand and discuss further this part.
Some relevant examples
Mo, X., Ge, W., Ren, Y., Zhao, D., Wei, D., & Romano, D. (2024). Locust-Inspired Jumping Mechanism Design and Improvement Based on Takeoff Stability. Journal of Mechanisms and Robotics, 16(6), 061013. https://doi.org/10.1115/1.4063406
Mo, X., Ge, W., Miraglia, M., Inglese, F., Zhao, D., Stefanini, C., & Romano, D. (2020). Jumping locomotion strategies: From animals to bioinspired robots. Applied Sciences, 10(23), 8607. https://doi.org/10.3390/app10238607
Sayyad, A.; Seth, B.; Seshu, P. Single-legged hopping robotics research—A review. Robotica 2007, 25, 587–613
Kram, R.; Dawson, T.J. Energetics and biomechanics of locomotion by red kangaroos (Macropus rufus). Comp. Biochem. Physiol. Part B Biochem. Mol. Biol. 1998, 120, 41–49.
Armour, R.; Paskins, K.; Bowyer, A.; Vincent, J.; Megill, W. Jumping robots: A biomimetic solution to locomotion across rough terrain. Bioinspir. Biomim. 2007, 2, S65.
Consider rephrasing the opening sentences for better engagement. For example, "With its insistence on the 'survival of the fittest', the biological structure of nature has become more and more reasonable" could be reworded for clarity and impact.
Add transitional sentences or phrases to better connect different sections of the introduction.
Clearly state the research problem and the specific objectives of the study earlier in the introduction.
If using acronyms such as CFD, consider defining them upon first use to ensure that readers unfamiliar with the terms can follow the text without confusion.
Explain why real-time observation is difficult and how researchers are addressing these challenges.
For instance, it could be useful to include and comment on works using different strategies to investigate jumping animals
Some sentences are complex and may benefit from simplification for improved readability.
Ensure consistent use of terminology throughout the introduction to avoid confusion.
Provide additional details about the function of each skeletal element in facilitating jumping.
Provide a more detailed explanation of the joint degrees of freedom for both forelimbs and hind limbs.
Discuss more the specific functions of muscles involved in hind limb extension and recovery (e.g., semimembranosus, gluteus, biceps femoris, gastrocnemius).
Enhance the explanation of the anatomical analysis of frogs and how it helps observe the range of motion in each joint.
Provide a more detailed explanation of the trajectory model of frog jumping motion. Discuss the significance of different stages (take-off, flight, landing) and how the frog's body posture changes during these stages.
How are the evaluation criteria for frog jumping performance, such as ground reaction force, joint torque, and jumping speed, determined?
Authors mention the need for a simplified bone structure model due to technical limitations. Consider providing a more detailed discussion on the specific technical challenges faced and potential solutions or future advancements that could overcome these limitations.
The table listing the structural model dimensions and mass is helpful. However, it would be beneficial to provide more insight into how these dimensions were determined or if they are based on existing frog anatomy studies.
Clarify the units used for dimensions and mass. Ensure consistency and clarity in units throughout the document.
The description of the simulation setup is detailed, but more information on the choice of parameters, such as the squat angle and the reasoning behind the specific joint angles applied during the simulation, would enhance the clarity of the methodology.
Consider including additional performance metrics or analyses to evaluate the accuracy of the simulation in replicating frog jumping behavior. This could involve comparing simulation results with observed frog jumping data.
The section effectively compares different structural models with varying degrees of freedom. However, it would be valuable to discuss how these simplifications impact the accuracy of the simulation compared to real frog movements.
The discussion of simulation results need a deeper interpretation of the implications of the observed trends or discrepancies between models could enhance the understanding of the biomechanics involved.
Expand the section on potential future improvements or refinements to the modeling approach. This could involve advancements in technology or methodologies that might address current limitations.
A deep English revision is needed.
Comments on the Quality of English Language
See previous comments.
Author Response
Thank you for your affirmation and valuable comments of this manuscript.
- Thank you for your careful review and valuable comments on the introduction. Firstly, we connect different parts of the introduction by adding transition sentences and phrases. Then, the sentence is modified to further expand and discuss the research status of jumping robots, clarify the specific objectives of the research problems, and improve the clarity and coherence of the expression to enhance the overall readability of the manuscript;
In addition, it is worth noting that the focus of this manuscript focuses on the structural modeling analysis of the frog-inspired jumping robot. Therefore, we focus on the analysis of the research status of the frog-inspired jumping robot in the introduction, and summarize the existing advantages and shortcomings, which more reflects the importance and necessity of multi-structure model analysis to select the appropriate structure for the research of the frog-inspired jumping robot. Then, the research methods of frog structure and motion mechanism are analyzed, and the time-saving, labor-saving and convenient simulation method is selected to analyze the influence of frog structure model on motion performance. The last paragraph briefly describes the structure of this manuscript. If you still feel that the quality needs further improvement, please give us another opportunity to revise and improve it.
- Thank you for your suggestion, we defined it when we first used abbreviations to ensure that readers unfamiliar with the term can follow the text without confusion.
- We explain the statement that it is difficult to obtain frog motion information by real-time observation. This is because the frog 's jumping movement is short and explosive. The general photographic camera cannot effectively and clearly capture the frog 's jumping movement. Even after the whole jumping movement is captured by the high-speed camera, it is necessary to obtain the frog's limb movement information through a series of modeling analysis, so real-time observation is very difficult. Researchers should establish appropriate coordinate system transformation on the basis of frog movement to obtain more accurate motion information.
- Thank you for your suggestions on the analysis of the movement mechanism of frogs. We further strengthened the explanation of the anatomical analysis of frogs, discussed the specific functions of hindlimb muscles during stretching and recovery, and provide a more detailed explanation of the joint freedom of limbs. At the same time, we also analyzed the trajectory model of frog jumping movement, discussed the significance of different stages and the attitude change of frog, and determined the evaluation criteria of frog jumping performance based on this.
- Thank you for your suggestions in detail. The size and quality of the model were determined based on the anatomical study of the actual frog, and the units of size(mm) and quality(g) were defined to ensure the consistency and clarity of the units in the entire document.
- Thank you for your suggestions on the simulation results. Indeed, as you said, we compared the structural models of different degrees of freedom. In order to enhance the understanding of the biomechanics involved, the meaning of the differences between our models was explained in more depth. At the same time, the modeling method that needs to be improved and perfected is explained, which points out the direction for future optimization research.
- Thank you for your suggestions on language. We have carefully read and revised the language of the manuscript. Please review it again. Thank you again for the time and effort you have put into improving this manuscript.
If there are any more questions, please don’t hesitate to contact us. We would be grateful if the manuscript could be considered for publication in Biomimetics. Thanks again for your time and detailed comments from the reviewers.

Reviewer 2 Report
Comments and Suggestions for Authors
In this paper, simulation analysis results of frog-inspired jumping performance were presented. Followings are comments for this paper.
1. The analysis results of jumping performance are explained using graphs, but it is necessary to present the actual jumping operation implemented. If the jumping motion is presented using captured images as shown in Figure 3, readers will be able to more intuitively understand the differences in results depending on the proposed model.
2. Figure 8 shows the trajectory of each joint for Model 1, but the results for Models 2, 3, and 4 are not presented in this paper. An explanation is needed as to why only the results for one model are presented.
Author Response
Thank you for your valuable comments on improving the quality of this manuscript.
- Thank you for this good suggestion, we have added a Figure 9 similar to Figure 3 to present the jump motion, so that readers can intuitively understand the motion effect of the model. Since the difference between the motion performance of different models is reflected in the motion distance and speed, we only added the motion effect diagram of Model 1 to avoid causing a repeated feeling to the reader. I hope you can agree with this. Thank you for your valuable comments.
- Indeed, as you said, Figure 8 shows the trajectory of each joint of Model 1. However, models 2, 3 and 4 can be obtained in this figure, because these three models are based on the degree of freedom simplification of model 1. Model 2 simplifies the internal and external rotation of the hip joint on the basis of Model 1. Model 3 simplifies the flexion and extension of the ankle joint on the basis of Model 2, and Model 4 further simplifies the abduction and adduction of the hip joint on the basis of Model 3. According to the principle of the control variable method, model 2,3 and 4 are consistent with model 1 except for the trajectory of the simplified joint. Therefore, we can get the joint trajectories of these three models from Figure 8.

Reviewer 3 Report
Comments and Suggestions for Authors
Authors have studied, modelled and simulated frog jumping using different models with different degrees of freedom. They show the models closest to the real world. They claim that this conclusion will provide the basis for a frog-inspired robot.
The state of the art should be updated with some more recent references, there are a several studies about frog-inspired jumping robots and simulation (ie: https://journals.sagepub.com/doi/10.1177/1687814018782303). I recommend a table describing the different published models about frog jump. Explain the main purpose and applicability of this study.
In relation with dynamic models, it is needed to improve their definition, even it would be advisable to extend the model including the full movement of the frog jump including landing.
About the text:
· Authors should indicate the friction coefficient used (flipper-ground) and justifying the used value. If authors used a contact model, please, define it in detail.
· A list of dynamic properties (table): mass, inertia properties of each body of the simulated models is required.
· When frog models jump, each joint could be as a pendulum. Have authors model a damping or constraints in each joint? Please describe in detail. During take-off, front limbs are moving free?
· The movement is not clearly explained. The initial movement is leaded by tarsometatarsal joint movement?. Describe how authors model this movement and its basis.
· Have authors added gravity force to the model? The Velocity (i.e. Figure 9) reaches a constant value of 1000 mm/s. It is necessary to explain.
Author claim that it is necessary to establish a mode (lines 236-238) for modelling several actions including landing, but only study the take-off movement.
Figure 6 should be improved from readability point of view. There are texts superposed.
Discussion should include comparison with the results of other authors.
Conclusion section should include limitations of the study (of the models), but also the future works.
Authors didn’t explain nor cite previous papers, also for discussion section:
· Fan J, Du Q, Dong Z, Zhao J, Xu T. Design of the Jump Mechanism for a Biomimetic Robotic Frog. Biomimetics. 2022; 7(4):142. https://doi.org/10.3390/biomimetics7040142
· https://doi.org/10.1016/S1672-6529(08)60023-2 (Figure 1 and 3 came from this paper, cite it and indicate the differences with the present paper)
Author Response
Thank you for your valuable comments on improving the quality of this manuscript.
- Thank you for the recent literature. We have carefully read and analyzed these literatures in detail and summarized the characteristics of different frog jumping models. On the basis of comparative analysis, the main purpose and applicability of this study are explained, and relevant literature is cited. Thank you again for the literature.
- Thank you for your suggestions on the dynamic model. We have improved its definition and included the whole process of frog jumping including landing. In order to ensure that the structural model can take off smoothly, we add contact between the limbs and the ground. The static friction coefficient is 0.7, and the rationality of the use value is explained. Meanwhile, the weight, length of each limb and the joint torques in the simulation process are listed in Table 1 and Table 2 respectively.
- Each joint in the model adds different constraints according to the different degrees of freedom during modeling. The jump motion is driven by the spline curve function generated according to the motion trajectory shown in Figure 8. The forelimb plays the role of adjusting the movement posture during the take-off process, and recovers to both sides of the torso after it leaves the ground. Therefore, the forelimb also has a certain law of motion, not free movement.
- Thank you for your valuable comments. We further elaborated the movement process of the frog to make the reader clearer. At the same time, the range of motion angle and degree of freedom of each joint can be known from the analysis of musculoskeletal characteristics, which is the basis for the driving sequence of each joint of the model.
- Thanks to your careful review. We added gravity to the simulation process to imitate the actual motion environment. However, the speed value in Figure 10 is actually not a constant value. The frog model entered the flight stage from about 0.23s. During the very short period of 0.02s from 0.23s to 0.25s, the speed of the model did not decrease much, so it seems to be a constant from the figure. It can also be seen from the jump distance of each model that the speed will gradually decrease under the action of gravity and enter the landing stage. Thank you again for your careful review.
- Indeed, as we have said, it is necessary to develop a model to simulate multiple actions including landing. In fact, it can be seen from the newly added Figure 9 that the model we established can achieve a variety of actions including landing through the movement of each joint. However, since the force and speed of the take-off stage determine the performance of the entire jump movement, we focus on the analysis of the influence of different joint degrees of freedom on the performance of the movement through the skeletal model, and then select the structural model with the best jump performance. At the same time, the landing action, stability determination and simulation research are also one of the follow-up research works. In the conclusion part, we summarize the limitations of this study and the future work direction.
- Thank you for your comments on Figure 6. Fig.6 is a virtual prototype of the frog skeleton structure model established in the simulation software. The text part only shows the coordinates of the points with constraints (all at the center of each joint). Readability has no effect on the understanding of the modeling process. The most important thing is that the added motion pair is readable so that the reader can understand the motion relationship of the joint. In addition, the model is in the contraction state before the take-off, and the distance between the limbs is relatively small, which makes the text coordinate overlap of the constraint points unavoidable, but this will not affect the reader 's understanding of the modeling process and the content of the manuscript. Thanks again for your careful review.
- Thank you for your suggestions for the discussion and conclusion. We added a comparative analysis with other research results, and cited the relevant research results in the references. On this basis, the limitations of this study are summarized, which points out the direction for future work.
- Thank you for your valuable comments. The analysis of frog musculoskeletal and its movement process is the basis for bionic modeling, so we conduct citation analysis on this basis. The difference is that the cited article establishes a simplified model based on the analysis of frog movement, focusing on a single model analysis. This manuscript is based on the establishment of a model with similar functions to frogs, and further compares and analyzes the influence of limb freedom on motion performance, and selects the optimal structural model while reducing the driving unit.
If there are any more questions, please don’t hesitate to contact us. We would be grateful if the manuscript could be considered for publication in Biomimetics. Thanks again for your time and detailed comments from the reviewers.

Round 2
Reviewer 1 Report
Comments and Suggestions for Authors
Authors addressed several of my comments, and the manuscript is improved.
Some crucial issue should be considered.
I would suggest authors to introduce more experimental approaches, and works using artefact beyond simulations.
Also, it would be interesting to comment in the discussion additional aspect affecting jumping performances in biological systems, that can be used to improve engineered artificial systems, wuch as lateralization, or different impact of age, etc.
Some example
Romano, D., Bloemberg, J., Tannous, M., & Stefanini, C. (2020). Impact of aging and cognitive mechanisms on high-speed motor activation patterns: evidence from an orthoptera-robot interaction. IEEE Transactions on Medical Robotics and Bionics, 2(2), 292-296. DOI: 10.1109/TMRB.2020.2977003
Lippolis, G., Bisazza, A., Rogers, L. J., & Vallortigara, G. (2002). Lateralisation of predator avoidance responses in three species of toads. Laterality: Asymmetries of Body, Brain and Cognition, 7(2), 163-183.
Comments on the Quality of English Language
English need a deep revision.
Reviewer 2 Report
Comments and Suggestions for Authors
1. Although the authors have responded to the reviewers' comments, there are still some important improvements that need to be made. In particular, explaining jumping performance only using graphs limits readers' understanding, so it is necessary to present images that can confirm performance changes according to the model.
2. Table 1 shows the sizes and weights of the proposed models, and it may be possible to perform simulations with these models. However, in an actual robot, a driving device must be included, and it is expected to be very difficult to implement this level of size and weight.
3. In Figures 10 to 13, the graph sizes are all different, and in Figures 14 and 15, the graph sizes are also different. It seems that more careful editing is needed.
4. In Figures 14 and 15, it is difficult to distinguish between model 1 and model 4, so it is necessary to change the color of the lines.
Reviewer 3 Report
Comments and Suggestions for Authors
Authors have addressed some of the proposed comments. Although there are some improvements included in the revised manuscript, I recommend to extend the research previous to publish the results.
Comments for improving the paper:
· Authors should justify the friction coefficient used of 0.7. If authors used a contact model, please, define it in detail.
· A list of dynamic properties (table): mass, inertia properties of each body of the simulated models is required.
· Have authors added gravity force to the model? The Velocity (i.e. Figure 9) reaches a constant value of 1000 mm/s. It is necessary to explain.
· Now, authors claim about the importance of take-off. In this case, I suggest the title of the paper should be adapted.
· Figure 6 should be improved from readability point of view. There are texts superposed. Not changed.
· Discussion should include comparison with the results of other authors. Not compared with.